# Glycosaminoglycans Modulate the Angiogenic Ability of Type I Collagen-Based Scaffolds by Acting on Vascular Network Remodeling and Maturation

**DOI:** 10.3390/bioengineering11050423

**Published:** 2024-04-25

**Authors:** Enrica Raffaella Grazia Salvante, Anca Voichita Popoiu, Amulya K. Saxena, Tudor Alexandru Popoiu, Eugen Sorin Boia, Anca Maria Cimpean, Florina Stefania Rus, Florica Ramona Dorobantu, Monica Chis

**Affiliations:** 1Doctoral School, Victor Babes University of Medicine and Pharmacy Timisoara, 300041 Timisoara, Romania; icamcsalvante@gmail.com (E.R.G.S.);; 2Emergency Hospital for Children Louis Turcanu, 300011 Timisoara, Romania; apopoiu@umft.ro (A.V.P.); boia.eugen@umft.ro (E.S.B.); 3Center of Expertise for Rare Vascular Disease in Children, Louis Turcanu Children Hospital, 300011 Timisoara, Romania; 4Department of Pediatric Surgery, Chelsea Children’s Hospital, Chelsea and Westminster Hospital NHS Fdn Trust, Imperial College London, London SW10 9NH, UK; amulya.saxena@nhs.net; 5Department III of Functional Sciences, Discipline of Medical Informatics and Biostatistics, “Victor Babes” University of Medicine and Pharmacy, No. 2 Eftimie Murgu Square, 300041 Timisoara, Romania; 6Department of Microscopic Morphology/Histology, Victor Babes University of Medicine and Pharmacy, 300041 Timisoara, Romania; 7National Institute of Research for Electrochemistry and Condensed Matter, Aurel Paunescu Podeanu Street 144, 300569 Timisoara, Romania; rusflorinastefania@gmail.com; 8Department of Neonatology, Faculty of Medicine and Pharmacy, University of Oradea, 410001 Oradea, Romania; 9Department ME2/Rheumatology, Rehabilitation, Physical Medicine and Balneology, Faculty of Medicine, George Emil Palade University of Medicine, Pharmacy, Science, and Technology of Târgu Mureş, 540088 Targu Mures, Romania; monicacopotoiu@gmail.com; 10Clinic of Rheumatology, Emergency County Hospital of Târgu Mureş, 540088 Targu Mures, Romania

**Keywords:** collagen scaffolds, neoangiogenesis, IKOSA app, choriollantoic membrane (CAM)

## Abstract

Type I collagen, prevalent in the extracellular matrix, is biocompatible and crucial for tissue engineering and wound healing, including angiogenesis and vascular maturation/stabilization as required processes of newly formed tissue constructs or regeneration. Sometimes, improper vascularization causes unexpected outcomes. Vascularization failure may be caused by extracellular matrix collagen and non-collagen components heterogeneously. This study compares the angiogenic potential of collagen type I-based scaffolds and collagen type I/glycosaminoglycans scaffolds by using the chick embryo chorioallantoic membrane (CAM) model and IKOSA digital image analysis. Two clinically used biomaterials, Xenoderm (containing type I collagen derived from decellularized porcine extracellular matrix) and a dual-layer collagen sponge (DLC, with a biphasic composition of type I collagen combined with glycosaminoglycans) were tested for their ability to induce new vascular network formation. The AI-based IKOSA app enhanced the research by calculating from stereomicroscopic images angiogenic parameters such as total vascular area, branching sites, vessel length, and vascular thickness. The study confirmed that Xenoderm caused a fast angiogenic response and substantial vascular growth, but was unable to mature the vascular network. DLC scaffold, in turn, produced a slower angiogenic response, but a more steady and organic vascular maturation and stabilization. This research can improve collagen-based knowledge by better assessing angiogenesis processes. DLC may be preferable to Xenoderm or other materials for functional neovascularization, according to the findings.

## 1. Introduction

Collagen is the most prevalent protein in the animal kingdom’s extracellular matrix and belongs to the fibrous protein family. Due to its high biocompatibility, collagen is an ideal biomaterial for implantable medical devices and scaffolds for in vitro testing. Different collagen-based solutions are being created and manufactured, including matrices, porous sponges, membranes, and threads for surgical and dental applications [1].

Moreover, collagen matrices have been historically known as promising biomaterials for a wide range of medical applications, particularly in tissue engineering and regeneration [2]. Collagen and materials derived from collagen have been used in medical applications for more than 50 years. In the past decade, there has been a significant increase in scholarly articles investigating the use of collagen in tissue engineering scaffolds. Collagen materials, such as soluble collagen injections, solid scaffolds, patches, and decellularized collagen matrices, have great potential in treating chronic wounds, burns, venous and diabetic ulcers. They also have versatility in various medical fields including plastic-reconstructive surgery, general surgery, urology, proctology, gynecology, ophthalmology, otolaryngology, neurosurgery, dentistry, cardiovascular surgery, bone and cartilage surgery, and cosmetology pathologies [3]. Their unique properties, including biocompatibility, biodegradability, and the ability to mimic the natural extracellular matrix (ECM), make them suitable for various regenerative therapies. Moreover, in dentistry, collagen matrices have gained significant attention in bone tissue regeneration, providing a scaffold for cell attachment, proliferation, and differentiation [4]. Additionally, their hemostatic properties make them valuable as surgical hemostatic agents, effectively controlling bleeding and promoting wound healing [5,6].

Xenoderm and dual-layer collagen solutions stand out, among the various collagen matrices available, for their specific applications and clinical outcomes. Xenoderm, derived from decellularized porcine ECM, has demonstrated efficacy as a wound-healing adjuvant therapy, particularly in treating burn injuries. Its porous structure and ability to absorb wound exudates promote granulation tissue formation and accelerate healing [7,8,9]. Moreover, wound healing efficacy is related to the idea that the collagen ECM clusters and promotes adhesion, proliferation, and differentiation at the cellular level. In the study by Matouskova et al., porcine dermal patches were demonstrated as an efficient solution in regenerative wound healing, as they promote keratinocyte proliferation and stratification [8].

Dual-layer collagen sponges, on the other hand, have found their main niche in dental surgery and, more generally, in bone regeneration procedures. Their biphasic (in layers) composition, with a dense and a porous layer, provides both structural support and a favorable environment for cell growth and differentiation, facilitating bone regeneration [10]. Additionally, dual-layer collagen matrices show superior performances even when compared to autogenous bone grafts, especially when it concerns bone regeneration. The layers of DLC offer mechanical stability, replicating the structural integrity of natural tissues. This mechanical support is crucial for providing a stable scaffold for bone formation and preventing bone collapse. The dual-layer structure of the DLC, made out of collagen and glycosaminoglycans, facilitates cell infiltration allowing osteoprogenitor cells to migrate into the scaffold and differentiate into osteoblasts, the bone-forming cells [11,12,13,14,15,16].

Despite the widespread use of these collagen matrices, there remains a paucity of data on the critical factor governing their effectiveness in regenerative medicine: vascularization [17]. Neoangiogenesis, the formation of new blood vessels, is essential for wound healing and tissue regeneration, providing oxygen and nutrients to the regenerating tissues [10,18].

The formation of the vascular network is mostly determined by the structure of the surrounding microenvironment, and is greatly influenced by the unique characteristics of blood flow. Existing evidence demonstrates that both DLC and Xenoderm possess a significant capacity for vascularization. However, there is currently no published data about their angiogenic potential and their capability to rebuild blood vessels. To address this gap in knowledge, we conducted a comparative study of Xenoderm and dual-layer collagen matrices in the chick chorioallantoic membrane (CAM) model. The CAM model offers a simple, cost-effective, and bioethically sound platform for studying angiogenesis in vivo [19,20,21].

Our study aimed to investigate de novo the neovascularization potential of Xenoderm and dual-layer collagen matrices in a comparative setting.

In the present study, we will focus not only on angiogenic potential of both collagen scaffolds, but also primarily on vascular remodeling assessed by the quantification of vascular loops and vascular network formation. Understanding the relative angiogenic properties of these matrices is crucial for assessing their scientific validity and optimizing their use in regenerative medicine applications.

## 2. Materials and Methods

### 2.1. Material Selection and Material Grafting into the Chick Embryo CAM

In order to compare the effects of Xenoderm and DLC on CAM vascularization, it was compulsory to implant the two biomaterials onto the surface of the chorioallantoic membrane of the chick embryo. Therefore, the integration with the living tissue system was enhanced by means of artificial intelligence (IKOSA App) to confirm whether and how the grafted biomaterials induced angiogenesis and lastly, functional vascular network formation. Different materials were needed to perform the experiment. The materials are listed as follows:

Xenoderm (Helix Pharma, Pondicherry, India) is a dermal substitute composed of a collagen-based matrix. The collagen matrix is derived decellularized porcine dermis and is crosslinked to create a stable scaffold. Xenoderm is used in clinical practice to treat a variety of skin conditions, including burns, chronic wounds, and diabetic ulcers.

Dual-layer collagen scaffold (DLC) (BIOPAD, Salt Lake City, UT, USA) is a biocompatible scaffold made of two layers. One layer is made of collagen type I, which promotes cell adhesion and proliferation, while the second layer is composed of glycosaminoglycans (GAGs).

Additional materials used for the realization of the experiment included 70% concentrated alcohol, ParaPlast sealing tape, 10% buffered formalin solution, and paraffin. A stereomicroscope was used to visualize the microscopic sections.

The chorioallantoic membrane assay preparation involved the initial step of select-ing 60 fertilized hen eggs using in ovo transillumination to determine the presence of developing embryos. Subsequently, the eggshells were cleansed using 70% concentrated alcohol, and then the eggs were incubated for a duration of 72 h at a temperature of 37 °C in an environment with a humidity level of 60%. On the fourth day of incubation, a puncture was made on the narrow end of each egg to extract roughly three milliliters of egg albumen, using ParaPlast to close the puncture. On the next day, the experiment included the establishment of a shell window to evaluate the integrity of the chorioallantoic membrane and the viability of the embryo. One of the two types of biomaterials was placed onto the CAM. The specimens were separated into two groups. The first group, consisting of 10 eggs, got small pieces of Xenoderm as implants. The second group, also consisting of 10 eggs, was implanted with small pieces of DLC as grafts. The biomaterials were applied directly onto the CAM surface in a non-invasive manner after lightly scratching it. The experiment concluded on the 13th day of incubation.

### 2.2. IKOSA CAM Assay and Network Formation Assay

To accurately assess the angiogenetic processes onto the CAM brought by the two different implanted biomaterials, we performed an automated analysis using the IKOSA app (KML Vision, Graz, Austria). The software is an artificial intelligence-enhanced app and includes a multitude of features for the rapid, accurate, but complex evaluation of different experimental models. For the present study, two IKOSA applications have been selected: the CAM assay (version 3.1.0) and the network formation assay (version 2.1.0).

The CAM assay allowed us to measure the total area of the vascular network, vessel length, vessel thickness, and the number of branching points. In addition to these parameters, we also considered parameters to evaluate the functional morphology of the vascularization, such as tube length, and number of tubes/tube length.

The network formation assay enabled the vascular loops analysis.

After exporting the data from the Ikosa app in .xls format, we conducted the sta-tistical analysis using the XLSTAT (version 2022.4.5.). Several graphs were created to illustrate the patterns of various variables over the three observation periods.

We employed a stereomicroscope to observe the stained slices. We obtained photographs of the segments using a digital camera. Subsequently, we employed ImageJ software (version 1.54) to scrutinize the photos and precisely measure the various properties.

## 3. Results

This study aimed to assess and compare the impacts of Xenoderm and DLC on CAM vascularization. Our research revealed that both biomaterials resulted in a notable stimulation of blood vessel growth and the development of a functional network of blood vessels. These findings suggest that DLC may be more effective at promoting vascular network formation, as well as vascular functionality, compared to Xenoderm (Figure 1).

The following parameters have been selected: (1) The variable “roi_size” represents the total number of pixels in the region of interest that was analysed; (2) vessels_total_area [Px^2]: the total sum of the areas of all the vessels that have been detected in the image, measured in square pixels; (3) vessels_num_branching_points: the count of the points where the discovered blood vessels divide into smaller branches; (4) covered area [Px^2]: the total area of all cells or tubes in the image, measured in pixels; (5) num_tubes: all the tubes that have been identified to be patent; (6) total_tube_length [Px]: the sum of the lengths of all tubes in the image, measured in pixels; (7) area [Px^2]: the area estimated within the loop, measured in square pixels; (8) perimeter [Px]: the measurement of the total length of the loop’s circumference. In order to assess the level of integration between the two materials and the live tissue model, stereo-microscopic images were captured at specific time points (days 1, 3, and 5). Subsequent analysis of the recorded data was then conducted. The region of interest (ROI) for the DLC was 17,056,325 squared pixels, while the ROI for the Xenoderm was 1,443,520 squared pixels. In order to facilitate a comparison of the observed values, they were divided by the ROI and thereafter expressed as percentages.

### 3.1. CAM Assay

The Ikosa app permits the evaluation of the vascularization through the feature “CAM assay”. The following images, tables, and graphs compare the two materials as integrating onto the CAM surface after grafting. Figure 1 compares stereomicroscopic images with CAM assay analysis of the two different biomaterials at different time periods.

#### 3.1.1. Vessel Total Area/ROI

The value “vessel total area/ROI” is depicted in the following Table 1 and represented in the Figure 2A.

The vessel_total_area/ROI exhibits a gradual and natural growth from day 1 to day 3. Xenoderm has rapid growth from day 0 to day 3, followed by a decline from 32.10% to 16.30%. In contrast, DLC exhibits slower growth and a less sudden decrease of just 3.86%. This suggests that DLC promotes angiogenesis and vascularization in a sustainable and more efficient manner. The data pertaining to Xenoderm indicate a significant increase from day 1 to day 3, but the subsequent drop implies a less seamless integration with the living model.

#### 3.1.2. Vessels_Num_Branching_Points/ROI

This variable, presented in Table 1 and Figure 2B, represents the percentage of new vascular ramifications in relation to the region of interest (ROI). The percentage of DLC vessels branching points increases from day 1 to day 5 slowly but constantly, indicating that DLC promotes neoangiogenesis and vascular branching over time. The trend exhibits a consistent rise during all three observations. On the other hand, Xenoderm percentages show an increase from the first to the second observation, followed by a stagnation, with little change in the last observational period.

#### 3.1.3. Vessels_Number_Branching_Points/Vessels_Total_Area

This variable, represented in Figure 1C and Table 1, is the ratio of the number of branching points to the total area of vessels and it is a quantitative measure of the complexity of the vascular network’s branching structure. A greater ratio signifies a network that is more intricate and extensively branched. The data for the DLC and Xenoderm indicate that Xenoderm exhibits greater branching complexity than the DLC at all three time points (Table 1 and Figure 2C). Over time, the DLC exhibits a decreasing tendency in the ratio of the number of branching sites to the total area of vessels. These findings indicate that the DLC vascular network has a decrease in branching ramifications as time progresses. The Xenoderm trend indicates that the ratio of vessels_number_branching_points to vessels_total_area increases with time. This indicates that the Xenoderm vascular network grows increasingly intricate yet less mature.

#### 3.1.4. Total_Tube_Length [Px]/Covered_Area

The ratio of total tube length over covered area is a measure of the density of the vascular network. A higher ratio indicates a denser vascular network. This means that there are more vessels per unit area of tissue. The data show that Xenoderm has a higher ratio of total tube length over the covered area than the DLC during the two first observations, and then DLC has a higher value on day 5. The trend for both Xenoderm and DLC is that the ratio of total tube length over the covered area decreases from day 1 to day 3, and then increases from day 3 to day 5. This suggests that the vascular network becomes less dense initially, and then denser over time. This may be due to the fact that the tissues are still developing and growing on day 1, and the vascular network needs to adapt to the changing needs of the tissues. (Figure 2D). The increase rate of the ratio of total tube length over covered area can be calculated using the following formula:Increase rate = (Value at day 5 − Value at day 3)/Value at day 3 × 100%

Using this formula, we can calculate the following increase rates for the Xenoderm and the DLC:Xenoderm: (4.01% − 3.27%)/3.27% × 100% = 22.63%
DLC: (4.04% − 1.10%)/1.10% × 100% = 272.73%

This shows that the DLC has a much higher increase rate than the Xenoderm. This means that the vascular network in the DLC is becoming denser at a much faster rate than the vascular network in the Xenoderm. The data are depicted in Table 1.

#### 3.1.5. Number_Tubes/Total_Tube_Length [Px]

The ratio of number of tubes to total tube length is a complex measure that can be influenced by a variety of factors, including the maturity of the vascular network, the presence of remodeling, and the specific tissue type. In a developing vascular network, the ratio of number of tubes to total tube length may be high. This finding demonstrates an immaturity of the circulatory system as the vessels are shorter and less developed. However, over time, the vessels may mature and elongate, leading to a decrease in the ratio of number of tubes to total tube length. Additionally, remodeling of the vascular network can lead to the removal of inefficient vessels and the formation of more efficient ones. This can also lead to oscillations in the ratio of number of tubes to total tube length. Therefore, a lower ratio of number of tubes to total tube length indicates that the vascular network is more mature and efficient. Overall, the ratio of number of tubes to total tube length is a useful tool for assessing the functionality of the vascular network, but it is important to consider all of the relevant factors when interpreting the results. In Table 1 and Figure 2E, while the values for DLC remain constant and then slightly decrease (3.29%, 3.35%, 3.05%), the parameter for Xenoderm shows a rise from 2.81% at day 1 to 5.40% at day 5 (Table 1 and Figure 2E).

#### 3.1.6. Vascular_Mean_Thickness

Another powerful means of calculating vascular functionality is by considering the “vessels_mean_thickness.” (Figure 2F and Table 2). The thickness of the of the vascular walls’ structures serves as a robust indicator to assess the maturity of each vascular entity, reflecting their functionality and capacity to facilitate and impulse blood flow. The mean thickness of the vessels found in the implanted eggs exhibits distinct trends, depending on the type of implant. The mean thickness of vessels in Xenoderm-implanted eggs demonstrates a decreasing trend over time, reducing from 11.98 Px on day 1 to 10.10 Px on day 5, suggesting that the Xenoderm implant does not promote an increase in vessel robustness over time. Conversely, the mean thickness of vessels in DLC-implanted eggs exhibits an increasing trend, rising from 25.12 Px on day 1 to 120.18 Px on day 5. Comparing the mean thickness of vessels in Xenoderm- and DLC-implanted eggs indicates that the Xenoderm implant does not effectively support the healthy development of vessels. In general, thicker vessels tend to be more elastic, enabling them to expand and contract more effectively in response to changes in blood pressure. This makes thicker blood vessels more efficient at transporting blood and oxygen to tissues. Overall, the assessment of mean thickness provides valuable insights into the vascular functionality and the impact of different implants on vessel development.

### 3.2. Vascular Loops Analysis by Network Formation Assay (NFA)

The vascular loop analysis performed with the network formation assay of the IKOSA app showed that the egg implanted with Xenoderm developed one vascular loop in during day 1, three vascular loops at day 3, and six vascular loops at day 5. The average loop area was 1552 P^2, 1702.67 P^2, and 7697 P^2, respectively. Showing an increase of 490% from day 1 to day 5. This is compatible with the idea of vascular maturity and vascular networking increasing more every day, and creating more complex loops and vascular networks every day (Figure 3).

The loop analysis performed by the IKOSA network formation assay demonstrated that DLC exhibits an even larger capability for forming vascular loops. In particular, after one day from the implant, DLC showed the formation of four vascular loops, with an average of 471,430 P^2 area each and an average perimeter of 2598.955 P. During the second observation, which happened during day 3, the loops observed were 14 with an average area of 30,350 P^2 and a perimeter of 513 pixels. Nonetheless, in the last day of observation, for a much smaller ROI of DLC, two loops were observed, having an average perimeter of 184.5 P and an average area 2196 P^2.

### 3.3. Comparative Statistical Analysis

A statistical comparison between DLC and Xenoderm related to the main angiogenic parameters evaluated by IKOSA was carried out. Statistical analysis of angiogenic parameters evaluated for Xenoderm showed no statistically significant correlation between branching points, vascular mean thickness, vascular area, and tube number/total tube length. By contrast, strong statistically significant correlations have been found for DLC (Table 3) between branching points/vascular mean thickness (*p* = 0.005), and branching points/total tube length per covered area (*p* = 0.005), but not for vascular total area. The strongest statistical significance observed was vascular mean thickness related to total tube length/covered area (*p* < 0.001). This correlation confirms and sustains DLC’s ability to induce a strong angiogenic response, but also to promote proper vessel remodeling and maturation.

## 4. Discussion

Collagen is a highly adaptable biomaterial that has significant importance in various practical fields. It is widely used in alimentary sciences and in the pharmaceutical industry. Additionally, collagen is utilized in the manufacturing of sports accessories like tennis rackets [11,22]. Collagen is the most abundant fibrous protein and makes up most of the extracellular matrix (ECM) in all species [23,24]. Collagen is a prevalent component in human tissues and exhibits various significant attributes, including cell recognition signals, the capacity to form 3D scaffolds of different configurations, manageable mechanical qualities, and natural degradation abilities. These qualities make collagen an ideal material for creating tissue-engineered scaffolds for a wide range of medical purposes. The appeal of collagen as a biomaterial mostly relies on its status as a naturally abundant component of the extracellular matrix. Consequently, it is regarded as an inherent part of the body, rather than an external substance [24]. The constituents of collagen engage in a sequential manner with one another, and with other elements of the extracellular matrix (ECM), to form structures with varying degrees of interconnection and specific functions. Collagen is crucial for preserving the structural and biological properties of connective tissues. An extensive understanding of these features has enabled the development of novel biomaterials that mimic the structural and biological properties of natural tissues, particularly tissues primarily consisting of collagen type I, III, and IV [25,26,27].

Surprisingly, the intrinsic ability of collagen scaffolds to induce and promote angiogenesis is less reported in the literature, despite the fact that some collagen-based biomaterials (as Xenoderm and similar) are frequently used in clinical practice for skin wound healing [7,28,29] or other reconstructive procedures [15,30,31].

De novo collagen scaffolds’ angiogenic ability to recruit blood vessels in the absence of other cellular interactions is less studied in the literature. The architecture heterogeneity of collagen scaffolds represents one of the main drivers of their vascularization. Collagen fiber arrangements differentially guide the recruitment of blood vessels and their structural patterns.

The present study was focused on the study of the endogenous angiogenic ability of two types of collagen scaffolds frequently used in clinical practice, but less characterized related to their angiogenic potential: Xenoderm (a skin dermis-derived collagen scaffold) [32] and dual-layer collagen sponge (usually used for hemostasis) [33]. Both Xenoderm and DLC have not been studied before on chick embryo chorioallantoic membrane models, regarding their ability to acquire new blood vessels from the adjacent microenvironment. Moreover, for most papers related to the angiogenic potential of both scaffolds, there were applied conventional methods to assess angiogenesis as microvessel density or the assessment of growth factors from the GAGs component by chemical or molecular methods. The artificial intelligence based IKOSA platform allowed us to assess angiogenesis with tests able to assess not only the number and density of newly formed blood vessels, but also by evaluating the dynamic criteria of angiogenesis processes as number of branching points, total vessel length and total vessel area, vessel wall thickness, and vascular loops. These criteria dynamically characterize angiogenic processes from their early stages of vessels sprouting, to their full maturation.

Xenoderm contains type I porcine collagen, while dual-layer collagen is structured in two distinct layers: one containing type I collagen and another layer filled with glycosaminoglycans. Both type I collagen and glycosaminoglycans promote angiogenesis [18,34], but our experimental study demonstrated that the hybrid collagen scaffold induces a different dynamics of angiogenesis steps. Type I collagen from Xenoderm induced a rapid and potent angiogenic response. This finding was sustained by a rapid increase in branching points number, total tube length, and total vascular area for Xenoderm compared to DLC in a similar period. Despite an accelerated increase in vessel total area for Xenoderm compared to DLC on day 3 of experiment, the total vessel area at the end of the experiment was higher for DLC than for Xenoderm. This may be surprising, but when we correlated these data with the findings derived from the vascular loops analysis, we found that the total vascular area developed in the DLC model is higher compared to Xenoderm, most probably due to higher maturation and stabilization of blood vessels induced by GAGs. These data may be important for choosing proper collagen scaffolds to be used in different organs tissue engineering. Previous experimental findings using DLC as scaffold for tissue bioprinting reported similar results on dual-layer collagen scaffolds’ advantages as one of the most proper materials for tissue bioprinting [35], and also for tissue vascularization [18], most probably due to the presence of a high amount of glycosaminoglycans. Glycosaminoglycans (GAGs) are tissular components mandatory for vascular development [36]. GAGs highly modulate both endothelial cells migration, proliferation, and tube formation, but also vascular remodeling and maturation due to their effects on perivascular smooth muscles cells also [37,38]. Recently, Pilloni et al. demonstrated that the addition of hyaluronic acid did not enhance the formation of new blood vessels in the early steps of reparative gingival angiogenesis, but enhanced extracellular matrix remodeling and collagen maturation, facilitating an accelerated wound healing inside human gingiva [39]. These findings are in accordance with ours. Total tube length in the last day of the experiment was similar in between both scaffolds but vascular mean thickness exhibits a tremendous increase for DLC compared to Xenoderm. This means that DLC (most probably due to the presence of GAGs layer) has ability to induce a rapid maturation of newly formed blood vessels. Vessels’ maturation and remodeling by addition of perivascular smooth muscle cells is mandatory for a proper vascular function [40,41]. Conventional microscopic methods, such as as CD34/smooth muscle actin double immunostaining for endothelial/perivascular cells, respectively, may prove the presence of perivascular smooth muscle cells on tissue section, but they do not have enough sensitivity and specificity to appreciate differences related to vessel thickness variability due to the acquisition of smooth muscle cells. AI-based IKOSA CAM platforms, especially CAM assay and network formation assay tests, can evaluate by automated image analysis vessel thickness of the newly formed perfused blood vessels in vivo and correlate it with other angiogenic parameters. GAGs layer from the DLC structure induced significant differences related to angiogenesis steps. In 1986, Ausprunk described the GAGs distribution during chick embryo chorioallantoic membrane vessels development [42]. The author pointed out that sulphated GAGs help the maturation and stabilization of CAM vessels. Collagen glycosaminoglycan dual-scaffolds increase the recovery of brain lesions in the rodent model of brain lesions [43] (26) by stimulation of angiogenesis. The same team reported that collagen glycosaminoglycan dual-scaffold also seemed to play a critical role in promoting proliferation of perivascular cells, especially in the area surrounding the lesion (26). This is in accordance with our findings regarding vessels’ mean thickness for a similar combination in between collagen and GAGs.

## 5. Conclusions

In this work, we proved that a combination of type I collagen and glycosamino-glycans on a DLC scaffold can promote blood vessel remodeling and angiogenesis. On its own, type I collagen promoted angiogenesis’ initial stages, but did little to aid in the development, maturation, or stabilization of the newly formed vascular network. The results show that glycosaminoglycan and type I collagen mixtures are the best scaffolds for angiogenesis from tissue engineering, which means that a fully functional vascular network development may be sustained.

## Figures and Tables

**Figure 1 bioengineering-11-00423-f001:**
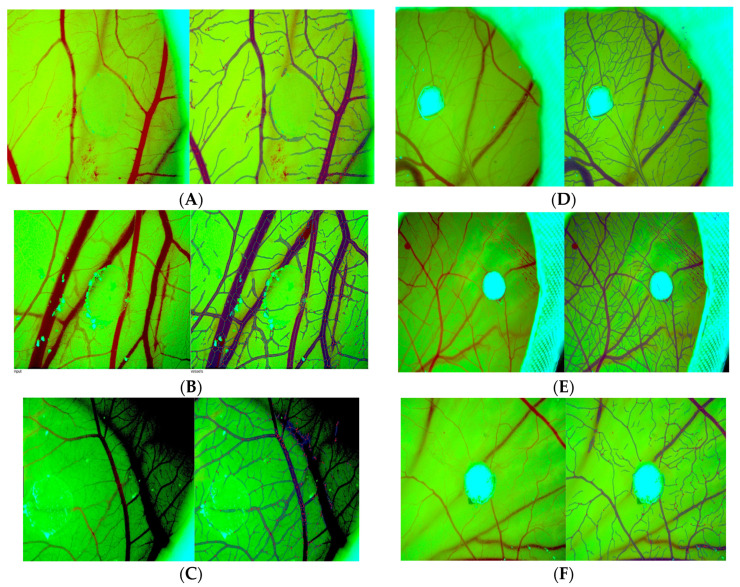
Dynamic evaluation of DLC (**A**–**C**) and Xenoderm (**D**–**F**) angiogenic potential by the assessment of branching points and vascular tubes morphometry through IKOSA app. CAM assay detected the branching points of new blood vessels by pointing them with red dots and automatically counted the branching points number.

**Figure 2 bioengineering-11-00423-f002:**
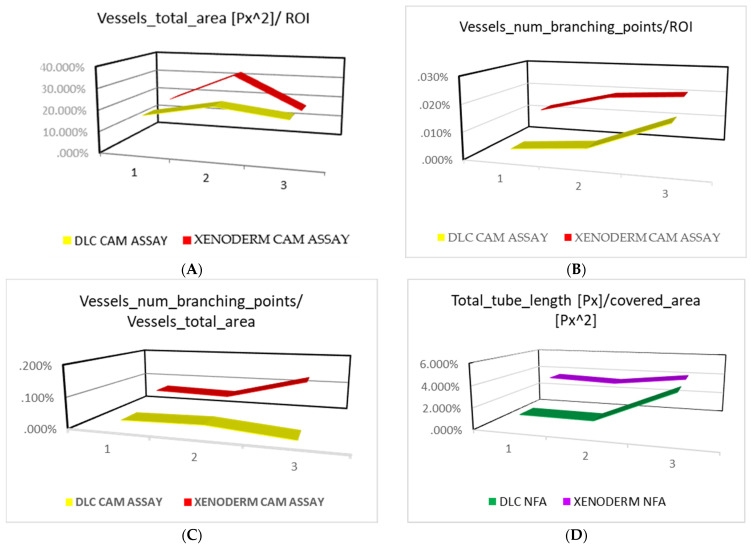
The value “vessel total area/ROI” is represented as a trend during the three observation periods (**A**). The second calculated variable is “vessel_num_branching_points/ROI” and it is represented as a trend during the three observation periods (**B**). The second calculated variable is “vessel_num_branching_points/vessels_total_area” and it is depicted as a trend (**C**). The number_tubes_length/Covered_area is depicted as a trend during the observation periods day 1, day 3, day 5 (**D**). The variable number_tubes/total_tube_length is depicted as a trend during the observation periods (**E**). Vascular mean thickness comparative assessment in between DLC and Xenoderm. Note the exponential increase of vascular thickness for DLC sustaining blood vessels maturation (**F**).

**Figure 3 bioengineering-11-00423-f003:**
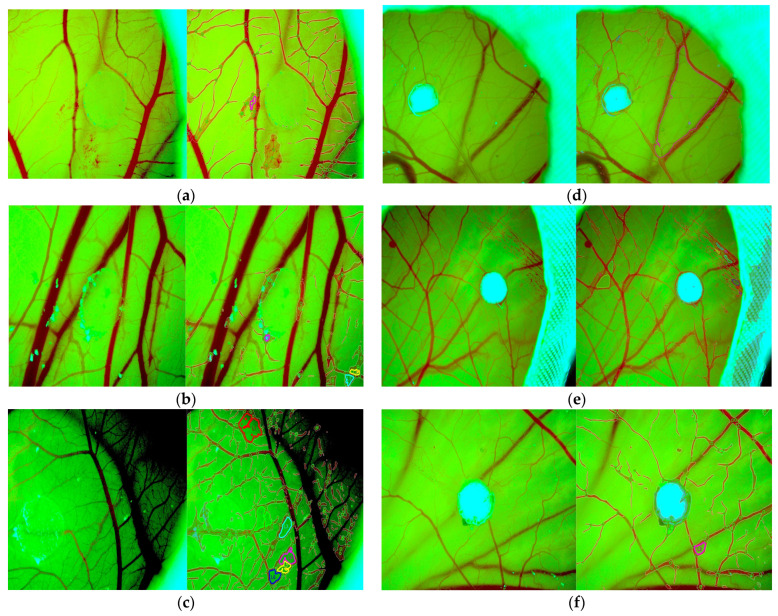
IKOSA network formation assay (IKOSA_NFA) evaluated the remodeling ability of newly formed vascular network (by assessing the vascular loops number and size), but also vascular mean thickness characterizing vessel maturation and stabilization. Significant differences have been found between DLC and Xenoderm. Vascular loop number, size, and density were significantly increased for DLC from day 1 (**a**) (one purple contour of one vascular loop) to day 3 (**b**) and day 5 (**c**) (six contoured loops in different colors which differentiate their size). For a similar period, for Xenoderm (**d**–**f**), there was a low number of vascular loops.

**Table 1 bioengineering-11-00423-t001:** CAM assay and network formation assay (NFA) parameters variability by comparative analysis in between dual-layer collagen (DLC, type I collagen + glycosaminoglicans) and Xenoderm (Type I collagen alone). The table compares the results of the DLC implanted eggs and the Xenoderm implanted eggs at the observational points: day 1 (1), day 3 (2), day 5 (3) for the following parameters: vessel_total_area [Px^2]/ROI; vessels_num_branching_points/ROI, meaning the number of ramification points encountered in the ROI and divided by the number of squared pixels encountered in the ROI; vessels_num_branching_points/vessels_total_area; vessels_num_branching_points/covered_area and number_tubes/total_tube_length [Px].

	DLC	Xenoderm
Parameteres	CAM Assay	CAM Assay
**Vessels_total_area** **[Px^2]/ROI**		
1	16.53%	18.29%
3	22.72%	32.10%
5	18.86%	16.30%
**Vessels_num_branching_points/ROI**		
1	0.00%	0.01%
2	0.01%	0.02%
3	0.02%	0.02%
**Vessels_num_branching_points/vessels_total_area**		
	0.02%	0.07%
	0.02%	0.06%
	0.00%	0.13%
	**NFA**	**NFA**
**Total_tube_length [Px]/covered_area [Px^2]**		
1	1.15%	3.53%
2	1.10%	3.27%
3	4.04%	4.01%
**Num_tubes/total_tube_length [Px]**		
1	3.29%	2.81%
2	3.35%	4.26%
3	3.05%	5.40%

**Table 2 bioengineering-11-00423-t002:** Vessels_mean_thickness [Px]. The table compares the results of the DLC-implanted eggs and the Xenoderm-implanted eggs at the observational points: day 1 (1), day 3 (2), day 5 (3).

Vessels_Mean_Thickness [Px]	DLC CAM Assay	Xenoderm CAM Assay
	25.12	11.98
	23.32	16.51
	120.18	10.10

**Table 3 bioengineering-11-00423-t003:** The table represents a statistical analysis through Pearson’s r, Spearmann’s rho and Kendall’s Tau B of the parameters calculated for the integration of dual-layer collagen (DLC).

		DLC_Vascular Mean Thickness	DLC_Branching Points
**DLC_branching points**	Pearson’s r	1.000 **	—
	*p*-value	0.005	—
	Spearman’s rho	0.866	—
	*p*-value	0.167	—
	Kendall’s Tau B	0.816	—
	*p*-value	0.110	—
**DLC_total tube length/covered area**	Pearson’s r	1.000 ***	1.000 **
	*p*-value	<0.001	0.005
	Spearman’s rho	1.000	0.866
	*p*-value	0.167	0.167
	Kendall’s Tau B	1.000	0.816
	*p*-value	0.167	0.110

Note. * *p* < 0.05, ** *p* < 0.01, *** *p* < 0.001, one-tailed.

## Data Availability

All results an data of this manuscript could be found to the main author of the present work.

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
