# Peer review of "Glycosaminoglycans Modulate the Angiogenic Ability of Type I Collagen-Based Scaffolds by Acting on Vascular Network Remodeling and Maturation"

_bioengineering, 2024, doi:10.3390/bioengineering11050423_

Round 1

Reviewer 1 Report

Comments and Suggestions for Authors

Thank you for the opportunity to review the manuscript. This study investigates the angiogenic potential of collagen type I-based scaffolds and collagen type I/glycosaminoglycans scaffolds using the chick embryo chorioallantoic membrane (CAM) model and IKOSA digital image analysis. Xenoderm showed rapid angiogenic response but incomplete vascular maturation, while the DLC scaffold induced slower yet more stable vascular maturation, indicating potential enhancement in collagen-based neovascularization.

1. I recommend including brand and company information of materials used.

2. The sections appear overly technical, and subtitles could enhance comprehension.

3. Combining data from Tables 1 to 5 into one table may align better with Figure 2.

4. Clarify the statistical analysis method—parametric or non-parametric? Was a normality test applied?

5. Tables 7 and 8 include unnecessary statistical data; focus on representative data for clarity.

6. The graphical abstract was not provided by the author.

Author Response

Response to REVIEWER 1

Dear Reviewer 1,

We are grateful and we would like to thank for your time and effort allocated for the review of our manuscript. Your comments and appreciation of our work were so valuable for us. We will respond point by point to all your suggestions.

Please see below our responses:

Thank you for the opportunity to review the manuscript. This study investigates the angiogenic potential of collagen type I-based scaffolds and collagen type I/glycosaminoglycans scaffolds using the chick embryo chorioallantoic membrane (CAM) model and IKOSA digital image analysis. Xenoderm showed rapid angiogenic response but incomplete vascular maturation, while the DLC scaffold induced slower yet more stable vascular maturation, indicating potential enhancement in collagen-based neovascularization.

  1. I recommend including brand and company information of materials used.

Done, thank you!

  1. The sections appear overly technical, and subtitles could enhance comprehension.

Revised, thanks!

  1. Combining data from Tables 1 to 5 into one table may align better with Figure 2.

Thank you for your suggestion. We combined all 5 tables into one table (Table 1) for both tests (CAM Assay and NFA) and the table was moved close to Figure 2.

  1. Clarify the statistical analysis method—parametric or non-parametric? Was a normality test applied?

Pearson’r test or Pearson’s correlation is the most common statistical test used to measure a linear correlation, or the linear relationship between two numerical variables. It does not assume normality although it assumes finite variance and covariance, and can be per extension considered a parametric test.

Spearman’s rho is a non-parametric test and can be use on ordinal data, it doesn’t assume normality, it measures the strength of association between the variables used.

Kendall’s coefficient is a correlation coefficient that doesn’t assume normality nor any other particular distribution, it can be used on ordinal and continuous variables, therefore is considered a non -parametric test.

  1. Tables 7 and 8 include unnecessary statistical data; focus on representative data for clarity.

Thank you for this valuable observation. Due to the lack of any significant correlation from parameters from table 7 we decided to remove table 7. A similar change was performed for table 8 where we removed the uneccesary data and we kept the significant statistical correlations only, significant for our study and relevant for DLC scaffold. By this change, the modified table 8 is now table 7. All changes were highlighted in red in the text of the manuscript.

  1. The graphical abstract was not provided by the author.

We made a graphical abstract and we uploaded it into the system!. Thank you for the suggestion.

Reviewer 2 Report

Comments and Suggestions for Authors

Tissue-engineered organs and implants hold promise for the replacement of damaged and diseased organs. With the help of scientifically based approaches, the concept of the role of glycosaminoglycans in modeling neoangiogenic effects was formed. I would like to note the high methodological level of work, the use of modern digital approaches in the presentation of research results.

Comments on the Quality of English Language

With the help of scientifically based approaches, the concept of the role of glycosaminoglycans in modeling neoangiogenic effects was formed. I would like to note the high methodological level of work, the use of modern digital approaches in the presentation of research results.

Author Response

Thank you for the positive remarks you mentioned about our experimental work. Hopefully , if it will be accepted, the present work could be helpful for other researchers who work in the field of bioengineering when they will choose the proper collagen scaffolds for the development of future in the lab organs.